# Medical Advances in Hepatitis D Therapy: Molecular Targets

**DOI:** 10.3390/ijms231810817

**Published:** 2022-09-16

**Authors:** Amelie Vogt, Sabrina Wohlfart, Stephan Urban, Walter Mier

**Affiliations:** 1Department of Nuclear Medicine, Heidelberg University Hospital, 69120 Heidelberg, Germany; 2Molecular Virology, Heidelberg University Hospital, 69120 Heidelberg, Germany

**Keywords:** hepatitis D, hepatitis B, farnesyl transferase, Hepcludex, viral entry mechanisms, antivirals

## Abstract

An approximate number of 250 million people worldwide are chronically infected with hepatitis B virus, making them susceptible to a coinfection with hepatitis D virus. The superinfection causes the most severe form of a viral hepatitis and thus drastically worsens the course of the disease. Until recently, the only available therapy consisted of interferon-α, only eligible for a minority of patients. In July 2020, the EMA granted Hepcludex conditional marketing authorization throughout the European Union. This first-in-class entry inhibitor offers the promise to prevent the spread in order to gain control and eventually participate in curing hepatitis B and D. Hepcludex is an example of how understanding the viral lifecycle can give rise to new therapy options. Sodium taurocholate co-transporting polypeptide, the virus receptor and the target of Hepcludex, and other targets of hepatitis D therapy currently researched are reviewed in this work. Farnesyltransferase inhibitors such as Lonafarnib, targeting another essential molecule in the HDV life cycle, represent a promising target for hepatitis D therapy. Farnesyltransferase attaches a farnesyl (isoprenyl) group to proteins carrying a C-terminal Ca1a2X (C: cysteine, a: aliphatic amino acid, X: C-terminal amino acid) motif like the large hepatitis D virus antigen. This modification enables the interaction of the HBV/HDV particle and the virus envelope proteins. Lonafarnib, which prevents this envelopment, has been tested in clinical trials. Targeting the lifecycle of the hepatitis B virus needs to be considered in hepatitis D therapy in order to cure a patient from both coexisting infections. Nucleic acid polymers target the hepatitis B lifecycle in a manner that is not yet understood. Understanding the possible targets of the hepatitis D virus therapy is inevitable for the improvement and development of a sufficient therapy that HDV patients are desperately in need of.

## 1. Introduction

Hepatitis D is the most severe form of viral hepatitis. According to estimates, around 12 million people worldwide suffer from hepatitis D virus (HDV) infection [1]. There may be many more undetected cases. Chronic hepatitis can lead to liver cirrhosis and carcinoma development. The HDV is a defective virus that depends on a preexisting hepatitis B virus (HBV) infection [2]. The current therapy consists of interferon-α, which induces a general antiviral reaction since there is no alternative available. Novel drugs are required to render therapy more efficient and to finally contain the disease. Currently, a wide range of drugs is under investigation and their targets and mode of action are reviewed in this work.

The hepatitis D virus is a small enveloped RNA virus of the delta virus genus [3,4]. The virus has a size of only 36 nm in diameter and its circular 1.7 kb RNA genome is single stranded (Figure 1) [4,5,6].

The open reading frame encodes for the hepatitis D antigen (HDAg), which is expressed in a large and a small form. The two antigens and the genome form the ribonucleoprotein (RNP), which is packed inside the virus envelope. HDV uses the hepatitis B virus envelope for packaging and entry. HDV is replicated solely by host enzymes which are in turn influenced by the hepatitis D antigens [7]. Eight genotypes of HDV have been identified to date, amongst which, only genotype one is spread worldwide [4,8]. The other genotypes are located in different regions (Figure 2) [9,10,11,12]. The course of the disease can differ greatly depending on the genotype [13].

## 2. Molecular Biology of HDV and HBV

During an HDV infection the infected cell produces, two viral proteins, the large and the small hepatitis D antigen (HDAg), perform different functions. While the small antigen increases genome replication, the large antigens promotes virus particle formation [14,15,16,17]. Therefore, the small antigen dominates in the early phase of cell infection whereas the large antigens is abundant in the later stages [18]. These proteins originate from the same open reading frame of the HDV genome, thus making them identical in major parts of their sequence. The large antigen is extended by 19 amino acids at the C-terminus, as described in the next paragraph. The RNA functions as a self-cleaving ribozyme that is also essential for replication [19]. Both antigens bind to genomic RNA. Several parts of the antigens are considered to participate in the RNA binding [20,21,22,23]. The large antigen connects the ribonucleoprotein (RNP) to the HBV envelope [15]. This interaction is essential for the assembly of an infectious particle. In contrast to the small antigen, the large antigen has a domain for binding to the hepatitis B surface antigen (HBsAg) [15,24]. The exact mechanism of this interaction is unclear but multiple studies reported the importance of the large HDV binding domain consisting of 19 C-terminal amino acids [2,25,26,27]. The domain is farnesylated at a cysteine residue [24]. Both the amino acid sequence and the farnesylation are inevitable for the protein–protein interaction [25]. Two subunits of the envelope proteins have been identified as crucial for the interaction of HBsAg and the large HDAg. The function of the cytosolic loop formed by the small HBsAg was investigated. It was found that mutation of amino acids 24–28 [27] and arginine at position 79 caused inhibition of virion formation [28]. Moreover, tryptophan residues at the C-terminus of HBsAg were found to be essential [29,30]. The main functions of the large HDAg are HBsAg binding via the carboxy terminus and trans inhibition mediated by an N-terminal domain [17,24].

HDV can only spread in patients with a pre-existing HBV infection. When compared to HDV, HBV is a more complex DNA virus infecting hepatocytes specifically [28]. The virus has a size of only 42 nm in diameter and contains a 3.2 kb partially double-stranded DNA genome [29,30]. Its genome encodes multiple antigens and a reverse transcriptase which is required for genome multiplication [28]. Additionally, the reverse transcriptase plays a role in virus particle formation by binding to the hepatitis B virus capsid and the genome [31]. The capsid is formed by the hepatitis B core antigen (HBcAg). The hepatitis B genome is converted into covalently closed circular dsDNA (cccDNA), and the DNA species is used for transcription. The cccDNA can interact with host factors and function as a minichromosome, a DNA species that is spared from degradation and therefore persists inside the cell [32,33].

The hepatitis B surface antigen (HBsAg) forms the envelope of HBV and HDV. Upon hepatitis B infection, HBsAg is expressed in three forms, referred to as large, medium and small HBsAg. HBsAg is an effective antigen for HBV vaccination [34]. The organotropism of the hepatitis viruses depends on hepatocyte-specific interactions during the cell entry. In the case of HBV, the HBsAg forms an antigenic loop that can interact with heparan sulfate proteoglycans (HSPG) [35,36]. The composition of HSPG is cell specific and they are located on the cell surface. This interaction is responsible for virus-cell approximation in the viral life cycle [36]. The subsequent cell entry is facilitated by the large HBsAg, a protein that is myristoylated at the amino terminal. This N-terminal domain is referred to as preS1domain, and it is unique to the large envelope protein [36]. The attached fatty acid functions as a membrane anchor that brings the virus in close approximation to the cell membrane [37]. Because of the interaction of the preS1 domain and the Na+-taurocholate cotransporting polypeptide (NTCP) on the surface of hepatocytes, the virus is able to enter the cell [30,38]. Consequently, the large HBsAg is inevitable for infection. Once HDV enters the cell, the envelope separates from the RNP and the latter travels into the nucleus via a nuclear location signal (Figure 3) [4]. Inside the nucleus the small antigen promotes genome replication. This is accomplished by redirecting the host DNA-dependent RNA polymerase to bind to the RNA genome of HDV and to multiply it [4]. The HDV genome forms a rod-like secondary structure due to GC base-pairing [39] and it is suggested that the double strands imitate the DNA substrate [40]. The host RNA polymerase performs a rolling-cycle mechanism which results in a genome multimer that is self-cleaved via its ribozyme activity. The resulting monomers are ligated to form their circular configuration [18]. The HDV genome has a negative polarity and migrates into the nucleolus where the replication intermediate is produced [41]. This intermediate is called the antigenome, it additionally functions as a template for mRNA production [7]. Both the genome and the antigenome are multiplied via the rolling-cycle mechanism and encode a ribozyme. After the antigenome leaves the nucleolus, an editing step leads to the elongation of the open reading frame. This alteration allows the production of two proteins from one open reading frame. The adenosine deaminase I converts the UAG stop codon into an UIG codon, which is then converted into UGG [42,43]. Therefore, the small antigen stops after 196 amino acids at the UAG triplet, whereas the large antigen contains a tryptophan at this position and contains 19 additional amino acids [44]. Since the small and the large antigen perform opposite tasks during the viral life cycle, this enzyme plays an important role in the transition from replication to virus assembly [45]. In summary, the antigenome fulfills three functions after its exit from the nucleolus. First, it acts as an intermediate for genome replication. The rolling-circle mechanism uses the antigenome as a template to produce genome multimers. Secondly, the unmodified antigenome is required as the RNA template for the small antigen mRNA. Thirdly, the edited antigenome serves as a template for the large antigen mRNA, making use of its elongated open reading frame. Following the mRNA translation, the two HDV antigens migrate back to the nucleus [46]. When linked with the HDV genome, they form the ribonucleoprotein that is found in the cytoplasm and the nucleus [47]. Both antigens are modified posttranslationally [7]. The farnesylation of the large antigen is essential for the subsequent packaging of the RNP into the surface proteins originating from the hepatitis B viruses. HBsAg is located outside the nucleus, in an early Golgi compartment [48,49,50]. While the exact details of HDV assembly are not yet unveiled, it is very likely to be comparable to the mechanism of HBV assembly as the envelope proteins are shared. Lastly, the assembled virions exit the cell via the Golgi apparatus [51].

## 3. Current Therapy

### 3.1. Cytokines

Patients diagnosed with hepatitis D are treated with pegylated (PEG) interferon-α if eligible. The cytokine has immunomodulatory and antiviral properties [52]. Interferon-α-induced gene expression gives rise to proteins that interfere with the viral life cycle. These proteins interfere with viral entry mechanisms [53,54], target the cccDNA (which is responsible for hepatitis B protein expression [55]), decrease viral gene expression [56], activate zinc finger-induced RNA decay [57,58], and impede the release of virions [59]. PEG-interferon therapy causes a broad range of side effects and relapses can occur after interferon-α therapy [60,61].

### 3.2. Reverse Transcriptase

The hepatitis B virus encodes a reverse transcriptase, which is essential for its replication and packaging [28]. In order to block the elongation of the emerging viral DNA, substrate analogues of this enzyme can be designed [62]. Nucleoside analogues are already used in HBV therapy [63]. Surprisingly, they now show significant improvement in the interferon therapy during HDV treatment in combination therapies [64,65,66]. Although the HDV is coated by the envelope proteins of hepatitis B, the effect of nucleoside analogous alone does not provide an efficient treatment. A possible explanation is the fact that they do not target the cccDNA or integrated DNA involved in the synthesis of the envelope proteins. Nonetheless, nucleoside analogues could be part of combination treatment consisting of multiple anti-HDV drugs since their effect on HBV has been proven. Therefore, they could handle HBV-mediated effects during hepatitis D therapy.

### 3.3. Sodium Taurocholate Co-Transporting Polypeptide

NTCP is a transmembrane transporter expressed on the basolateral membrane of hepatocytes [67]. The glycosylated phosphoprotein is the first member of the solute carrier family 10 (SLC10) [68,69,70]. NTCP forms nine transmembrane domains [71] that oligomerize in the phospholipid bilayer [38]. In healthy individuals the NTCP is required for the circulation of bile acids [70]. Using the sodium gradient, the transporter binds two sodium ions and one taurocholate molecule and moves them across the membrane [72]. Additional substrates include bile acid derivatives, steroid and thyroid hormones, and some xenobiotics [73,74].

In 2012, Yan and Zhong et al. identified NTCP as the receptor used for the entry of hepatitis B and D [30]. The structure of NTCP has not been determined yet. Therefore, the rational design of entry inhibitors is directed either to the viral proteins or the substrates of NTCP, in particular bile acids. The bile acid analogues are capable of inhibiting viral entry since the binding of preS1 interferes with bile acid transport [38]. The immunosuppressing drug cyclosporin A was reported to inhibit HBV and HDV entry by interacting directly with NTCP [75]. Based on the preS1 domain of the large HBsAg, the first-in-class entry inhibitor Hepcludex was developed. Hepcludex is an optimized alternative to the 47 amino terminal amino acids of the preS1 domain. It is acylated with myristic acid to match the viral structure. The binding site of Hepcludex and cyclosporin A overlap, as proven by their mutual binding competition [75]. Hepcludex is well tolerated and efficient in HBV/HDV-infected patients in monotherapy as well as in combination with interferon-α [76,77]. The first-in-class entry inhibitor offers the promise to prevent the spread in order to gain control and eventually cure HBV and HDV. In July 2020 the European Medicines Agency granted Hepcludex conditional marketing authorization throughout the European Union. Hepcludex was recently tested in long-term monotherapy treatment and determined save and efficient to decrease HDV-RNA and alanine aminotransferase levels [78]. Although previous studies reported only minor side effects of Hepcludex, a case of immediate-type hypersensitivity reaction and a case of allergic skin reaction were reported [79,80]. Nonetheless, both patients were able to continue Hepcludex treatment.

## 4. Potential Therapy Targets

### 4.1. Farnesyltransferase

Farnesyltransferase (FTase) is the enzyme that catalyzes the farnesylation of proteins. During this posttranslational modification, a C15 isoprenoid lipid (the farnesyl residue) is covalently attached to a cysteine residue of a protein [81]. The enzyme is composed of an alpha and a beta subunit [82]. Farnesyl pyrophosphate (FPP) is the activated farnesyl derivative used in this reaction. The ordered binding mechanism of FTase, is initiated by the binding of FPP [83]. Subsequently, the carboxy terminus of the target peptide binds to the catalytic domain of the enzyme. Although different parts of the peptide can interact with the enzyme [84], the main interaction is based on the Ca1a2X (C: cysteine, a: aliphatic amino acid, X: C-terminal amino acid) farnesylation motif [85]. Additionally, the activity of FTase depends on two ions. The beta subunit coordinates a zinc ion, found inside the binding pocket [86]. It coordinates the cysteine residue during the reaction and is therefore part of the peptide substrate binding structure [87]. In addition, millimolar magnesium concentrations are a prerequisite for FTase activity [88]. It is assumed that the magnesium ions stabilize a negative charge, possibly that of pyrophosphate [87]. The mechanism by which the enzyme catalyzes the thioether bond is not understood yet. FTase is in charge of farnesylation in human cells and therefore mediates the membrane association and protein–protein interaction of different proteins [89]. One important substrate of the enzyme is the rat sarcoma G-protein (RAS). RAS plays an important role in carcinogenesis. The farnesyltransferase is a cytosolic enzyme [89] and after its farnesylation, RAS is anchored in the endoplasmic reticulum (ER) membrane [89]. 

During the hepatitis D life cycle, farnesylation of the large antigen is inevitable for virus assembly. The 19 amino acids unique for the large antigen and the farnesyl residue are mandatory for the binding to the envelope proteins of HBV [25]. The small HBsAg protein is a transmembrane protein located at an early Golgi membrane [50]. It is therefore conceivable that the membrane association of the large antigen close to HBsAg influences the interaction of both proteins.

The large HDAg contains a Ca1a2X motif consisting of the carboxy terminal amino acids Cys-Arg-Pro-Gln [24]. The cysteine is coordinated by a zinc ion both in the substrate and the product complex [87]. The a1 amino acid does not form specific interactions that contribute to substrate specificity [84]. The amino acid a2 forms a hydrogen bond with the beta subunit [84]. For this position numerous amino acids are accepted by the enzyme, and proline is one of them [87]. The terminal amino acid X is important for the differentiation whether a peptide is farnesylated or geranylgeranylated, it interacts with the enzyme through hydrogen bonds [84]. A terminal glutamine has been found to cause preferential farnesylation [84,87]. The carboxyl residue of the peptide terminus forms hydrogen bonds with both subunits [84]. The interactions of the cysteine and the carboxy terminus sets the size limit to four amino acids for this peptide motif. FPP binds to a hydrophobic funnel shaped binding pocket where the pyrophosphate is coordinated by positive charges [87]. FTase can bind FPP but not geranylgeranyl pyrophosphate because of a tryptophan residue that blocks the space required for a fourth isoprene unit [87]. Upon binding of a new substrate, the product migrates to a different part of the enzyme [85]. Therefore, product and substrate compete for enzyme binding.

Farnesyltransferase inhibitors can block the substrate binding pocket or the exit space [85]. The inhibitors can mimic the peptide substrate or FPP [87]. Even though the drugs were originally developed for cancer treatment, these inhibitors are now tested in HDV therapy. Lonafarnib is an FTase inhibitor currently under investigation in HDV therapy. A proof-of-concept trial confirmed the anti-HDV effect of lonafarnib [90]. Later studies confirmed the efficiency of the drug in combination with ritonavir and interferon alpha as an all-oral therapy [91,92]. 

### 4.2. HBV Surface Antigen and Subviral Particles

Nucleic acid polymers (NAP) are antiviral polymers that are effective against HBV and HDV by targeting their shared envelope protein [93]. NAPs are phosphorothioate oligonucleotides with a polyanionic backbone. Their activity is defined by their length, and hence amphipathicity and hydrophobicity, not by their sequence [94]. In a hydrophobic microenvironment the backbone is uncharged and can participate in hydrophobic interactions [95]. A solvent exposed hydrophobic domain is not common in mammalians but in enveloped viruses [93]. NAPs bind to exposed amphipathic α-helices on viral fusion glycoproteins and could avert natural protein interactions or conformations [94,96]. 

The NAPs intrinsically migrate to the liver and are able to enter hepatocytes [94]. Their exact mode of action is not understood but multiple observations that allow first assumptions were reported. Both an entry inhibition and decrease of HBsAg secretion are possible modes of action [94]. The virus approaches the cell by interacting with HSPG. Nucleic acid polymers are suspected to interfere with this interaction [93]. During an infection, the surface antigens self-assemble into empty, non-infectious subviral particles (SVP). The SVPs are secreted into the bloodstream [97]. During a NAP treatment, infected patients showed a reduction of the serum marker HBsAg, possibly because NAP inhibits SVP secretion or assembly [94,98]. This inhibition could be selective for SVP or include the secretion of hepatitis D virions [93]. The clearance of HBsAg enables the host immune system to gain control over the infection [94,98]. The reason for this is the interference of HBsAg with multiple mechanisms of the adaptive and innate immune system [58,99,100,101]. Additional post entry effects of NAP are discussed because of the observed reduction of hepatitis D genomic RNA during treatment which is not connected to the suppressed HBsAg secretion [95]. 

REP 2139 is a nucleic acid polymer currently under clinical investigation. It is administered as a chelate complex with an excess of divalent ions to avoid the depletion of ions from the patient after administration [102]. A recent phase 2 study in patients with chronic HBV demonstrated the improvement of standard therapy by the addition of REP 2139 without additional adverse reactions [103].

### 4.3. Additional Targets

Beyond the entry inhibitors, farnesyl transferase inhibitors and nucleic acid polymers, tested in clinical trials, several other drugs have been shown to offer the potential to suppress hepatitis D. Small interfering RNA was designed to target hepatitis B mRNA and to reduce viral replication [104,105]. So-called core protein allosteric modulators interfere with regular capsid formation by allosterically binding to the hepatitis B core protein that forms the HBV capsid [106]. The small molecules activate core protein assembly which leads to falsely aggregated capsid proteins [106,107]. Some core protein allosteric modulators are already tested in phase I clinical trials [108]. Furthermore, carbamoyl-phosphate synthetase 2, aspartate transcarbamylase, and dihydroorotase (CAD) were identified as proteins that are important for HDV replication [109]. N-(phosphonoacetyl)-L-aspartic acid, an anti-cancer drug, inhibits CAD and could consequently also interfere with the HDV lifecycle [110]. HBV transcription could be hindered by the drug pevonedistat. Hepatitis B regulatory X protein causes degradation of structural maintenance of chromosomes (Smc) 5/6 complex, hence making the transcription of HBsAg from the extrachromosomal cccDNA possible [111,112]. Pevonedistat inhibits the degradation of the Smc 5/6 complex and as a result reduces transcription of hepatitis B proteins [113]. Lastly, interferon lambda is under investigation as a drug for the treatment of HBV/HDV infections due to its effects, which are similar to those induced by interferon-α [114].

## 5. Conclusions

The novel drugs for the treatment of HBV/HDV show promising results in vitro and in vivo. In contrast to interferon-α, they directly target the viral lifecycle. Most of the current developments are tested in combination with interferon-α, which will continue to be of importance in HDV therapy. A combinational therapy could be efficient for HDV therapy to avoid development of resistance and maximize efficacy. Since HDV uses host proteins of hepatitis B proteins for its replication, hepatitis D proteins rarely qualify as targets. In this work we discussed multiple promising possibilities to interfere with the HDV life cycle. Further testing is required to identify the optimal treatment for HDV.

## Figures and Tables

**Figure 1 ijms-23-10817-f001:**
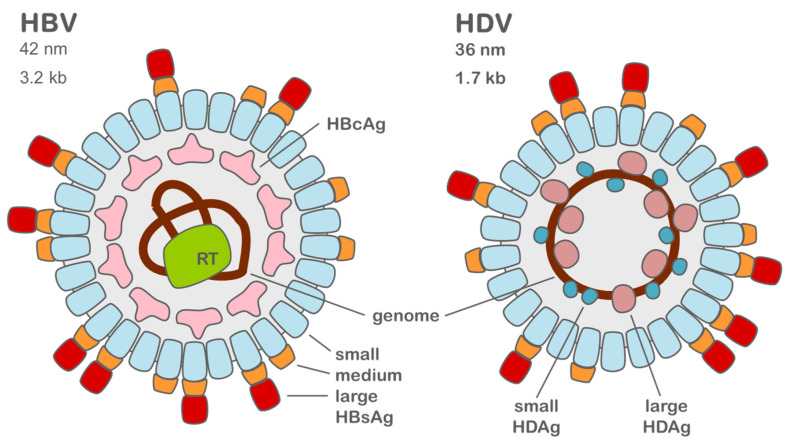
Comparison of HBV and HDV. Both viral envelopes consist of small, medium, and large HBsAg. HBcAg forms the HBV capsid which encloses the reverse transcriptase and the HBV genome. In the HDV particle, the envelope proteins enclose the ribonucleoprotein consisting of the genome and the small and large HDAg.

**Figure 2 ijms-23-10817-f002:**
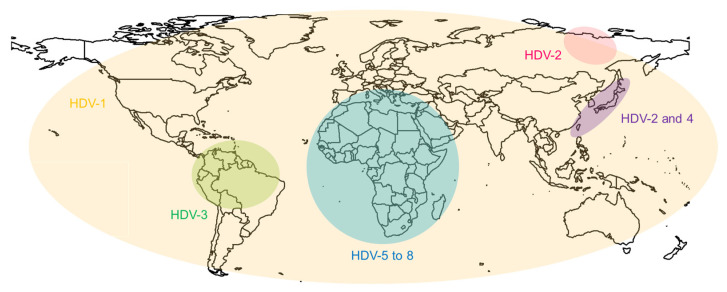
Global distribution of HDV genotypes. Genotype 1 is spread globally. Genotypes 2 and 4 are found in Taiwan and Japan. Genotype 2 is also found in parts of Russia. Genotype 3 dominates in the Amazon region. Genotypes 5 to 8 are located on the African continent.

**Figure 3 ijms-23-10817-f003:**
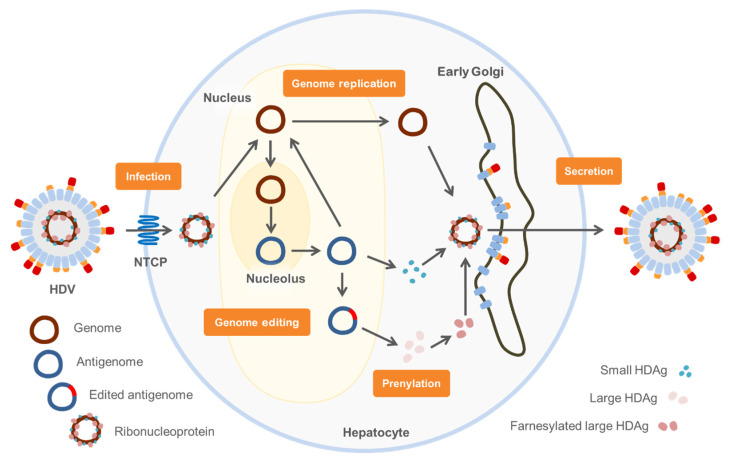
Hepatitis D virus life cycle. The hepatitis D virus enters hepatocytes via the NTCP. HBsAg and RNP separate and the genome migrates into the nucleolus. Inside the nucleus, several copies of the antigenome are produced from the genome via a rolling circle mechanism. These serve as a template for mRNA synthesis from which the small HDAg is translated. In addition, the antigenome is edited by adenosine deaminase I. The edited antigenome serves as a template for mRNA synthesis from which the large HDAg is translated. Subsequently, the large HDAg is farnesylated by the human farnesyltransferase. Several copies of the genome are produced from the antigenome via a rolling circle mechanism. Together with the two HDAg these form an RNP, which is enveloped by HBsAg in an early Golgi compartment and secreted.

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
