# Peer review of "Medical Advances in Hepatitis D Therapy: Molecular Targets"

_ijms, 2022, doi:10.3390/ijms231810817_

Round 1

Reviewer 1 Report

The manuscript “Medical Advances in Hepatitis D Therapy: Molecular Targets” reviews the current developments in HDV therapy, discussing the importance of studying the molecular biology of HDV and HBV to promising targets for antiviral therapy. In recent years, a number of reviews have been published on this topic. This review contains basic information on the HBV and HDV therapeutic targets and antivirals. However, to avoid repetition of already published data, it should be focused on very recent data.   

Major concerns:

The Introduction is too long and contains data on molecular biology of HDV and HBV that could be moved to separate section. Moreover, data presented in this section are fully represented in a large number of comprehensive reviews published in recent years. Thus, this part can be shortened significantly.

The structure of manuscript is not clear. For instance, section 2 is dedicated to current therapy, but nucleoside analogues that are currently used for HBV treatment are divided into separate section 3. It is advisable to group all data inti two major sections – Current therapy and Potential therapy targets with respective sub-sections.   

The data on new potential anti-HDV drugs presented in manuscript need a serious update.

Data on entry inhibitor (Hepcludex, or Bulevertide) should be updated with a number of recently published data on its use in the real world practice. For instance:  doi: 10.1111/apt.16945,  doi: 10.1016/j.jhep.2022.03.004, doi: 10.1016/j.jhep.2022.06.010, doi: 10.1111/liv.15330.

Data on farnesyltransferase inhibitor are also to be updated significantly, as the latest paper mentioned in this review is the phase 2A trial data published in 2015. Since then, a number of papers have been published on this topic (doi: 10.1002/hep.32259, doi: 10.1002/hep.29658)

Furthermore, the manuscript needs serious proofreading for the correct use of terms.

Minor comments:

lines 49-50: “As the course of the disease can differ 49 greatly, a connection to the performance of the different genotypes is discussed”. The term “performance” is not correct in this context.

- Figure 1. HDV genotype 2 is found not only in Taiwan and Japan. It is also prevalent in Yakutia (Russia). Please see the reference: Ivaniushina V, Radjef N, Alexeeva M, Gault E, Semenov S, Salhi M, Kiselev O, Dény P. Hepatitis delta virus genotypes I and II cocirculate in an endemic area of Yakutia, Russia. J Gen Virol. 2001 Nov;82(Pt 11):2709-2718. doi: 10.1099/0022-1317-82-11-2709.

- line 64: please provide the for explanation for RNP abbreviation as it is the first mention in the text.

- lines 72-77: language editing is recommended.

- line 82: “hepatitis capsid” – a typo?

- line 86: The term “relapse” is not correct in this context.

Author Response

Reviewer 1 Comments and Suggestions for Authors

The manuscript “Medical Advances in Hepatitis D Therapy: Molecular Targets” reviews the current developments in HDV therapy, discussing the importance of studying the molecular biology of HDV and HBV to promising targets for antiviral therapy. In recent years, a number of reviews have been published on this topic. This review contains basic information on the HBV and HDV therapeutic targets and antivirals. However, to avoid repetition of already published data, it should be focused on very recent data.

Thank you for this positive evaluation. The authors are grateful for the constructive comments, which have certainly allowed us to improve the quality of this manuscript.

major concerns

R1 #1   The Introduction is too long and contains data on molecular biology of HDV and HBV that could be moved to separate section. Moreover, data presented in this section are fully represented in a large number of comprehensive reviews published in recent years. Thus, this part can be shortened significantly.

The purpose of the comprehensive introduction was to conceive the basis to the subsequent description of the targets. Thus, this ends in the usual balancing act to prepare the reader with the fundamental information.

We hope that a in the revised manuscript, the issue could be solved by the proposed reconstruction using a separate section. In a second step, it would be possible to delete some of the most unambiguously published sections. We have identified and labeled these parts in grey. If you consider their deletion as an overall improvement, we are ready to delete them in accordance with your instructions.

R1 #2   The structure of manuscript is not clear. For instance, section 2 is dedicated to current therapy, but nucleoside analogues that are currently used for HBV treatment are divided into separate section 3. It is advisable to group all data into two major sections – Current therapy and Potential therapy targets with respective sub-sections.

We do agree, in fact this proposal makes more sense: the section headers were revised accordingly.

R1 #3   The data on new potential anti-HDV drugs presented in manuscript need a serious update.

Data on entry inhibitor (Hepcludex, or Bulevertide) should be updated with a number of recently published data on its use in the real world practice. For instance: doi: 10.1111/apt.16945, doi: 10.1016/j.jhep.2022.03.004, doi: 10.1016/j.jhep.2022.06.010, doi: 10.1111/liv.15330.

Data on farnesyltransferase inhibitor are also to be updated significantly, as the latest paper mentioned in this review is the phase 2A trial data published in 2015. Since then, a number of papers have been published on this topic (doi: 10.1002/hep.32259, doi: 10.1002/hep.29658).

Thank you for the provision of these interesting current references, we have included them in the revised manuscript.

R1 #4   Furthermore, the manuscript needs serious proofreading for the correct use of terms.

The manuscript was written to the best of our effort and knowledge. Honestly, we are not able to identify terms that are misused.

Minor comments

R1 #5   lines 49-50: “As the course of the disease can differ 49 greatly, a connection to the performance of the different genotypes is discussed”. The term “performance” is not correct in this context.

Thank you for your diligent corrections. In fact, this was a mistake which we corrected.

R1 #6   - Figure 1. HDV genotype 2 is found not only in Taiwan and Japan. It is also prevalent in Yakutia (Russia). Please see the reference: Ivaniushina V, Radjef N, Alexeeva M, Gault E, Semenov S, Salhi M, Kiselev O, Dény P. Hepatitis delta virus genotypes I and II cocirculate in an endemic area of Yakutia, Russia. J Gen Virol. 2001 Nov;82(Pt 11):2709-2718. doi: 10.1099/0022-1317-82-11-2709.

Again – we benefit from your thorough revision! The number of infected persons in Yakutia may be low, but as this information is available, we have amended the area in the figure.

R1 #6   - line 64: please provide the for explanation for RNP abbreviation as it is the first mention in the text.

The abbreviation is explained in the revised manuscript.

R1 #6   - lines 72-77: language editing is recommended.

The section was revised in accordance with your suggestion.

R1 #6   - line 82: “hepatitis capsid” – a typo?

This was corrected.

R1 #6   - line 86: The term “relapse” is not correct in this context.

The word “relapse” was deleted.

Reviewer 2 Report

Submitted manuscript reviews present advances and future developments in HDV therapy. The authors conclude that targeting the lifecycle of the HBV combined with IFN-alpha needs to be considered in HDV therapy. Understanding the possible targets of the HDV therapy is inevitable for the development of a sufficient HDV therapy. Hepcludex is an example of how understanding the viral lifecycle can give rise to new therapy options. Farnesyltransferase inhibitors like lonafarnib targeting another essential molecule in the HDV life cycle represent a promising target for hepatitis D therapy. HDV proteins rarely qualify as targets.

I have few concerns with this manuscript.

1.      Please insert references to Figures into text.

2.      Please change the order of the figure 1 and 2. The first figure should probably be the original Figure 2 inserted at the end of line 41 as Figure 1.

3.      The HDV size is 36 nm “in diameter” (l.42).

4.      Figure showing the global distribution of HDV genotypes should be referenced as Figure 2 on line 48.

5.      Please explain better the origin of 19 amino acids at the C-terminus of the large HDV antigen.

6.      Please indicate the size of HBV particle in text (l. 78).

7.      Figure “comparison of HBV and HDV”: please indicate the size of both particles and genomes in the Figure or in the legend to figure. Please explain why the graphical presentation of HBV genome inside virus particle seems to be linear.

8.      Please explain better „the three functions of the antigenome after its exit from the nucleolus“(l. 126).

Author Response

Reviewer 2 Comments and Suggestions for Authors

Submitted manuscript reviews present advances and future developments in HDV therapy. The authors conclude that targeting the lifecycle of the HBV combined with IFN-alpha needs to be considered in HDV therapy. Understanding the possible targets of the HDV therapy is inevitable for the development of a sufficient HDV therapy. Hepcludex is an example of how understanding the viral lifecycle can give rise to new therapy options. Farnesyltransferase inhibitors like lonafarnib targeting another essential molecule in the HDV life cycle represent a promising target for hepatitis D therapy. HDV proteins rarely qualify as targets.

Thank you for this positive evaluation. The authors are grateful for the constructive comments, which have certainly allowed us to improve the quality of this manuscript.

I have few concerns with this manuscript.

  1. Please insert references to Figures into text.

Thank you for this important hint, the figures are listed in the revised manuscript.

  1. Please change the order of the figure 1 and 2. The first figure should probably be the original Figure 2 inserted at the end of line 41 as Figure 1.

Figure 1 and 2 were exchanged in accordance with your suggestion.

  1. The HDV size is 36 nm “in diameter” (l.42).

The term “in diameter” was added.

  1. Figure showing the global distribution of HDV genotypes should be referenced as Figure 2 on line 48.

The figure was referenced as proposed.

  1. Please explain better the origin of 19 amino acids at the C-terminus of the large HDV antigen.

We hope that this information is provided in the manuscript (lines 123-127). Therefore, we have to be careful to avoid a second, redundant explanation. In the revised manuscript, we refer to the explanation in the respective paragraph.

  1. Please indicate the size of HBV particle in text (l. 78).

The size of HBV particles is provided in the revised manuscript.

  1. Figure “comparison of HBV and HDV”: please indicate the size of both particles and genomes in the Figure or in the legend to figure. Please explain why the graphical presentation of HBV genome inside virus particle seems to be linear.

The figure was revised in accordance with your suggestion.

  1. Please explain better „the three functions of the antigenome after its exit from the nucleolus“(l. 126).

The information on the function is restructured and improved in the revised manuscript.

Round 2

Reviewer 1 Report

The manuscript has been improved significantly and can be published in the current form. Authors addressed all comments in the revised manuscript. Parts labeled in grey can be kept in the text, as the changes in structure of the review have worked well.